**Data Availability Statement:** All relevant data are within the manuscript and its Supporting Information files.

**Funding:** The authors received no specific funding for this work.

# Impact of introducing a standardized nutrition protocol on very premature infants' growth and morbidity

Apolline Wittwer🆔*, Jean-Michel Hascoët

Department of Neonatology, Maternité Régionale, CHRU NANCY, DevAH University of Lorraine, France

* a.wittwer@chru-nancy.fr

## Abstract

### Background

Inappropriate nutritional intake in premature infants may be responsible for postnatal growth restriction (PGR) and adverse long-term outcomes.

### Objective

We evaluated the impact of an updated nutrition protocol on very premature infants' longitudinal growth and morbidity, and secondly the compliance to this new protocol.

### Design

All infants born between 26–32 weeks gestation (GA) were studied retrospectively during two 6-month periods before (group 1) and after (group 2) the introduction of an optimized nutrition protocol, in a longitudinal comparative analysis.

### Results

158 infants were included; 72 before and 86 after the introduction of the protocol (Group 1: (mean±SD) birthweight (BW) 1154±276 g, GA 29.0±1.4 weeks; Group 2: BW 1215±332 g, GA 28.9±1.7 weeks). We observed growth improvement in Group 2 more pronounced in males (weight z-score) at D42 (−1.688±0.758 vs. −1.370±0.762, p = 0.045), D49 (−1.696 ±0.776 vs. −1.370±0.718, p = 0.051), D56 (−1.748±0.855 vs. −1.392±0.737, p = 0.072), D63 (−1.885±0.832 vs. −1.336±0.779 p = 0.016), and D70 (−2.001±0.747 vs. −1.228±0.765 p = 0.004). There was no difference in females or in morbidities between the groups. We observed low compliance to the protocol in both groups: similar energy intake but higher lipid intake in Group 1 and higher protein intake in Group 2.

### Conclusion

The quality of nutritional care with a strictly-defined protocol may significantly improve weight gain for very preterm infants. As compliance remained low, an educational reinforcement is needed to prevent PGR.

**Competing interests:** The authors have declared that no competing interests exist.

**Abbreviations:** BPD, bronchopulmonary dysplasia; BW, birth weight; CLD, chronic lung disease; CVL, central venous line; DGOS, Direction générale de l'offre de soins; ESPGHAN, European Society of Paediatric Gastroenterology, Hepatology and Nutrition Committee; GA, gestational age; HM, human milk; IUGR, intra uterine growth restriction; MOM, mother's own milk; NEC, necrotizing enterocolitis; NICU, neonatal intensive unit care; PCA, post conceptual age; PDA, patent ductus arteriosus; PGR, postnatal growth restriction; PMA, post menstrual age; SD, standard deviation; WGA, weeks of gestational age.

## Clinical trial registration

This retrospective study was registered by ClinicalTrials.gov under number NCT03217045, and by the CNIL (Commission Nationale de l'Informatique et des Libertés) under study number R2015-1 for the Maternity of the CHRU of Nancy.

## Introduction

Inappropriate nutrition is an issue for premature infants as it may be responsible for postnatal growth restriction (PGR) and increased morbidity [1–4]. *Embleton* [2] demonstrated that 55% of PGR [5] were related to a cumulative deficit in energy and protein intake. Malnutrition in the first weeks of life is associated with short stature and adverse outcomes in adulthood [6–8]. The goal of nutrition is to allow body composition and outcomes similar to those of infants born at term [3,9]. Optimizing nutrition needs to start from birth as the window for PGR prevention and catch-up growth is rather narrow [10]. The prevention of nutrient deficits may be achieved through the implementation of optimized nutritional policies. More recent studies also demonstrated benefits on growth with the implementation of nutritional strategies [11–13].

We decided therefore to introduce a standardized nutritional protocol in our NICU and study outcomes on premature infants (including growth and morbidity).

In our unit, parenteral and enteral nutrition used to be adjusted according to the 2004 guidelines defining global nutritional intakes [14,15]. From May 1st, 2014, we decided to update our policy according to more recent ESPGHAN and other recommendations [16–19] available at the time of the study, and to follow a strict nutritional protocol rather than only global guidelines.

The objective of this study was to evaluate the longitudinal impact of implementing this strictly-defined nutrition protocol on very premature infants' growth and morbidity using a before/after comparison design, with a 6-months wash-out period. The secondary objective was to evaluate physicians' compliance to the standardized protocol.

## Patients and methods

### Study design

All infants born at the maternity hospital and hospitalized in our NICU were studied retrospectively during two 6-month periods, separated by a 6-month washout period, from May 1 to October 31, 2013 (group 1) then from May 1 to October 31, 2014 (group 2), which occurred before and after the introduction of an optimized nutrition protocol. We performed a longitudinal comparative analysis between these two independent groups. Collection of data from the infants' files was standardized.

### Methods

This retrospective study was registered by ClinicalTrials.gov under number NCT03217045, and by the French ethic committee "Commission Nationale de l'Informatique et des Libertés" (CNIL) under study number R2015-1 for the Maternity of the CHRU of Nancy. Parents' consent for using the collected clinical data of their infant was obtained and signed at admission.

Infants born between 26 to 32 weeks gestation (GA) and admitted to our NICU were included in the study. Infants who died before discharge or presented with any congenital

malformation were excluded. The primary outcome measure was to evaluate the impact of introducing a well-defined nutrition protocol on the longitudinal growth of the infants up to the time of discharge. Weight was assessed by daily measurements, every morning, as defined in our routine policy of care. To account for variations in gestational and postnatal age, body weight was converted into a z-score using the Olsen preterm infants' reference growth chart [20].

In order to account for the infants' size at birth, we also analyzed each point z-score difference from birth z-score. PGR was defined by a weight z-score <-1,5 DS before 36 post conceptual age (PCA) or discharge [21].

Secondary outcomes were the incidence of necrotizing enterocolitis (NEC) as defined by Bell [22]; interruption of nutrition and its duration; duration of parenteral nutrition; incidence of late onset sepsis; patent ductus arteriosus (PDA) requiring treatment; bronchopulmonary dysplasia (BPD) defined as the use of supplemental oxygen at 28 days of life; and chronic lung disease (CLD) defined as the need for supplemental oxygen at 36 weeks PMA. Antenatal steroids rate, ibuprofen, caffeine, and doxapram treatment were also recorded in the infants' files.

Finally, we evaluated weekly actual nutritional intake of the infants fed according to the guidelines compared to the new protocol to define the compliance of our team to the 2 policies. Enteral and parenteral nutrition were modified according to ESPGHAN recommendations. Proteins were prescribed at a higher rate in group 2, with a onset at birth, of 1.5 vs. 0.5 g/kg/d, and a goal of 3.5 vs. 2.5 g/kg/d. Lipids were prescribed earlier in group 2 and at higher rate of 1 g/kg/d on day 1 vs. 0.5 g/kg/d starting on day 2, with a goal of 3.5 vs. 3 g/kg/d. Enteral nutrition was initiated from day 1 in both groups but with an increase of 15–20 in group 2 vs.10-15 ml/kg/d in group 1 (Table 1).

Use of mother's own milk was promoted, but when it was not available banked donor's milk was used. Milk fortification was initiated from 80 ml/kg/d in both groups.

The parenteral solutions were prepared by the hospital pharmacy according to medical prescription.

In order to evaluate compliance to the protocol, energy, protein, lipid, carbohydrate, and sodium intake were reported weekly for each patient after collection of the raw data in the patient's individual daily file. Human milk content was assumed to be an average of 70 kcal/100 mL with 1.2 g protein, 4.2 g fat, and 7 g/100 mL carbohydrate[23]. Milk fortification and preterm infant formulas were also used and their formula compositions and nutritional content were based on the product labels.

Each intake (protein, carbohydrate, lipid) was compared to the theoretical recommendations of each nutritional protocol in use for the period. When the intake was out of the range of the recommendations, it was considered as not compliant to the protocol for the period.

Compliance to guidelines for each intake in the two groups was calculated by the average of compliant prescriptions compared with recommendations for each nutritional protocol in use for the period (in percentage +/- standard deviation).

## Statistics

Normally distributed data, assessed by a Shapiro-Wilk test of normality, are presented as mean values with SD; non-normally distributed data are presented as medians with the range of values. To evaluate differences between groups, we used the Student t test for continuous variables and the Chi$^2$ test or Fisher exact test when appropriate for categorical variables. For continuous variables not normally distributed, we used the Mann-Whitney U test. Categorical data are presented as actual numbers or percentages. Postnatal evolution of growth variables was assessed longitudinally by z-score. Observed differences were considered statistically

**Table 1. Old guidelines compared to new protocol.**

| | Day of life | D0 | D1 | D2 | D3 | D4 | D5 | D10 | D15 |
|---|---|---|---|---|---|---|---|---|---|
| **Fluids (mL/kg/d)** | **Old guidelines** | 70 | 90 | | | | | 160 | 160–180 |
| | **New protocol** | 80 | 90 | +10–20 | +10–20 | +10–20 | 130–160 | 160 | 160 |
| **Carbohydrates (g/kg/d)** | **Old guidelines** | 6–6,5 | **10–12** | | | | | **20–24** | |
| | **New protocol** | 6 | **8–10** | +1 | +1 | +1 | 12–14 | **18** | 18 |
| **Proteins (g/kg/d)** | **Old guidelines** | **0,5** | **0,5** | | | | 2 | **2** | 2–3.5 |
| | **New protocol** | 1,5 | 2 | +0,5 | +0,5 | +0,5 | 2,5 | **3,5** | 3,5 |
| **Lipids (g/kg/d)** | **Old guidelines** | 0 | 0,5 à 1 | | | | **1 to 2** | **2** | 2–3.5 |
| | **New protocol** | 0 | **1** | +0,5 | +0,5 | +0,5 | 3 | **3,5** | 3,5 |
| **Sodium (mEq/kg/d)** | **Old guidelines** | 0 | 1 | +0.5 | +0.5 | +0.5 | 1,5 to 2 | 3 to 4 | 3 to 4 |
| | **New protocol** | 0 | 1 | +1–2 | +1–2 | +1–2 | 2 to 4 | 4 | 4 |
| **Enteral Nutrition** | **Old guidelines** | 0 | 1–2 mL/kg/d | + 8–10 mL/kg/d | + 8–10 mL/kg/d | + 8–10 mL/kg/d | + 8–10 mL/kg/d | **100–150 mL/kg/d** | **150 mL/kg/d** |
| | **New protocol** | 0 | 6x 1 to 2 mL | +20 mL/kg/d | +20 mL/kg/d | +20 mL/kg/d | 80 mL/kg/d | **160 mL/kg/d** | **160 mL/kg/d** |
| | | | | | | | +1% Fortifier | **+3% Fortifier** | **+3% Fortifier** |
| **Energy (kcal/kg/d)** | **Old guidelines** | | 50 | | | | | **100** | |
| | **New protocol** | | 50 | | | | 90 | 120 | |

significant if $P < 0.05$. All analyses were performed with SYSTAT 13 software (2009, Systat Software Inc®, San Jose CA, USA).

## Results

### Demographic and clinical characteristics of the population

From May to October 2013, 85 eligible infants were included in the study. Thirteen were excluded: twelve died, and one file was lost. The remaining 72 infants (Group 1) had a mean birthweight (BW) of 1154 ± 276 g, and a mean GA of 29.0 ± 1.4 weeks. From May to October 2014, 90 eligible infants were included in the study. Four were excluded: two died and two files contained missing data. The remaining 86 infants (Group 2) had a mean BW of 1215 ± 332 g and a mean GA of 28.9 ± 1.7 weeks.

The clinical characteristics of the study population are displayed in Table 2. The groups were similar, with no significant differences for sex, birth weight, GA, prenatal steroids, PDA, or BPD. We observed a significant difference for PDA requiring surgery and for the duration of treatment with doxapram which did not impact the results (Table 2).

### Primary outcome: Impact on longitudinal raw weight and weight z-score

Postnatal weight loss occurred in both groups during the first week of life, with no difference in the minimal global weight on day 7 (1121±233 g in Group 1 vs. 1159±305 g in Group 2; p = 0.385). Mean weight loss was 2.8% in Group 1 and 4.9% in Group 2 (NS). There was no significant difference in raw weight between the two groups from birth to day 77. However, when looking at weight z-scores, we observed significant differences between the 2 periods from day 42 up to day 70 (Fig 1). Of note, z-score decrease was significantly smaller (about 0.5

**Table 2. Clinical characteristics and outcomes of the study population.**

|  | Group 1 (n = 72) | Group 2 (n = 86) | Total population (n = 158) | p |
|---|---|---|---|---|
| **Males, n (%)** | 37 (51.39) | 44 (51.16) | 81 (51.27) | 0.977 |
| **Mean birth weight (SD), g** | 1154 (±276) | 1215 (±332) |  | 0.21 |
| **Mean birth weight z-score (SD)** | −0.494 (±0.140) | −0.426 (±1.096) |  | 0.705 |
| **Mean gestational age (SD), weeks** | 29.0 (±1.4) | 28.9 (±1.7) |  | 0.69 |
| **Any prenatal steroids, n (%)** | 66 (91.67) | 76 (88.4) | 142 (89.9) | 0.632 |
| Complete course | 29 (40.3) | 38 (44.2) | 67 (42.4) | 0.266 |
| **PDA, n (%)** | 17 (23.6) | 13 (15.1) | 30 (19) | 0.187 |
| PDA surgically closed | 0 | 5 (5.8) | 5 (3.16) | 0.008 |
| PDA medically closed | 15 (20.8) | 13 (15.1) | 28 (17.7) | 0.705 |
| **Doxapram, n (%)** | 21 (29.2) | 28 (32.6) | 49 (31) | 0.611 |
| Mean duration, (SD) (days) | 10.8 (±11.54) | 22.89 (±11.73) |  | 0.001 |
| **BPD, n (%)** | 35 (48.6) | 35 (40.7) | 70 (44.3) | 0.7 |
| **CLD, n (%)** | 18 (25) | 20 (23.3) | 38 (24.1) | 0.622 |
| **NEC, n (%)** | 18 (25) | 14 (16.3) | 33 (20.9) | 0.17 |
| Stage 2 of Bell, n (%) | 13 (18) | 8 (9.3) | 21 (13.3) | 0.13 |
| Stage 2B of Bell, n (%) | 0 | 3 (3.5) | 3 (1.9) | 0.13 |
| Stage 3 of Bell, n (%) | 0 | 1 (1.2) | 1 (0.6) | 0.13 |
| **Alimentation withdrawal, n (%)** | 43 (59.7) | 42 (48.8) | 85 (53.8) | 0.196 |
| Duration of alimentation stop, days | 6.4 (±6.1) | 5.9 (±5.4) |  | 0.642 |
| **CVL, n (%)** | 70 (97.2) | 74 (86.0) | 144 (91.1) | 0.051 |
| CVL duration, days | 15.7 ±10.5 | 15.3 ±9.4 |  | 0.803 |
| **Infection, n (%)** | 32 (44.4) | 40 (46.5) | 72 (45.6) | 0.743 |
| **Parenteral alimentation duration, days** | 15.8 (±9.1) | 14.1 (±8.3) |  | 0.254 |
| **Hospitalization duration, days** | 54.9 (±27.1) | 51.5 (±31.7) |  | 0.314 |
| **Age at discharge, WGA** | 36.8 (±3.5) | 36.1 (±3.7) |  | 0.166 |

BPD: bronchopulmonary dysplasia; CLD: chronic lung disease; CVL: central venous line; NEC: necrotizing enterocolitis; PDA: patent ductus arteriosus; SD: standard deviation; WGA: weeks of gestational age.

z-score) and delayed in Group 1 as compared to Group 2 (day 70 vs. day 21, respectively) (Fig 1).

Looking at weight z-score evolution with regards to gender, we observed a significant difference between male and female infants: there was a significant difference in males from day 42 to day 70 (Fig 2A), but not in females (Fig 2B). However, when taking into account the weight z-score at birth, the analysis of each time point difference from birth was always above -1 for boys (Fig 3A) and without significant difference between groups or sex (Fig 3A and 3B).

## Secondary outcomes

We evaluated the potential impact of our new standardized protocol on population morbidity (Table 2). We evaluated the tolerance of enteral nutrition because of a faster increase in enteral nutrition volume, but found no difference between the two groups for the rate of NEC.

While our new protocol allowed for faster achievement of full enteral nutrition, there was no significant difference in the duration of hospitalization, central venous line and parenteral nutrition duration (which correspond with full enteral feeding), or the rates of infection (Table 2).

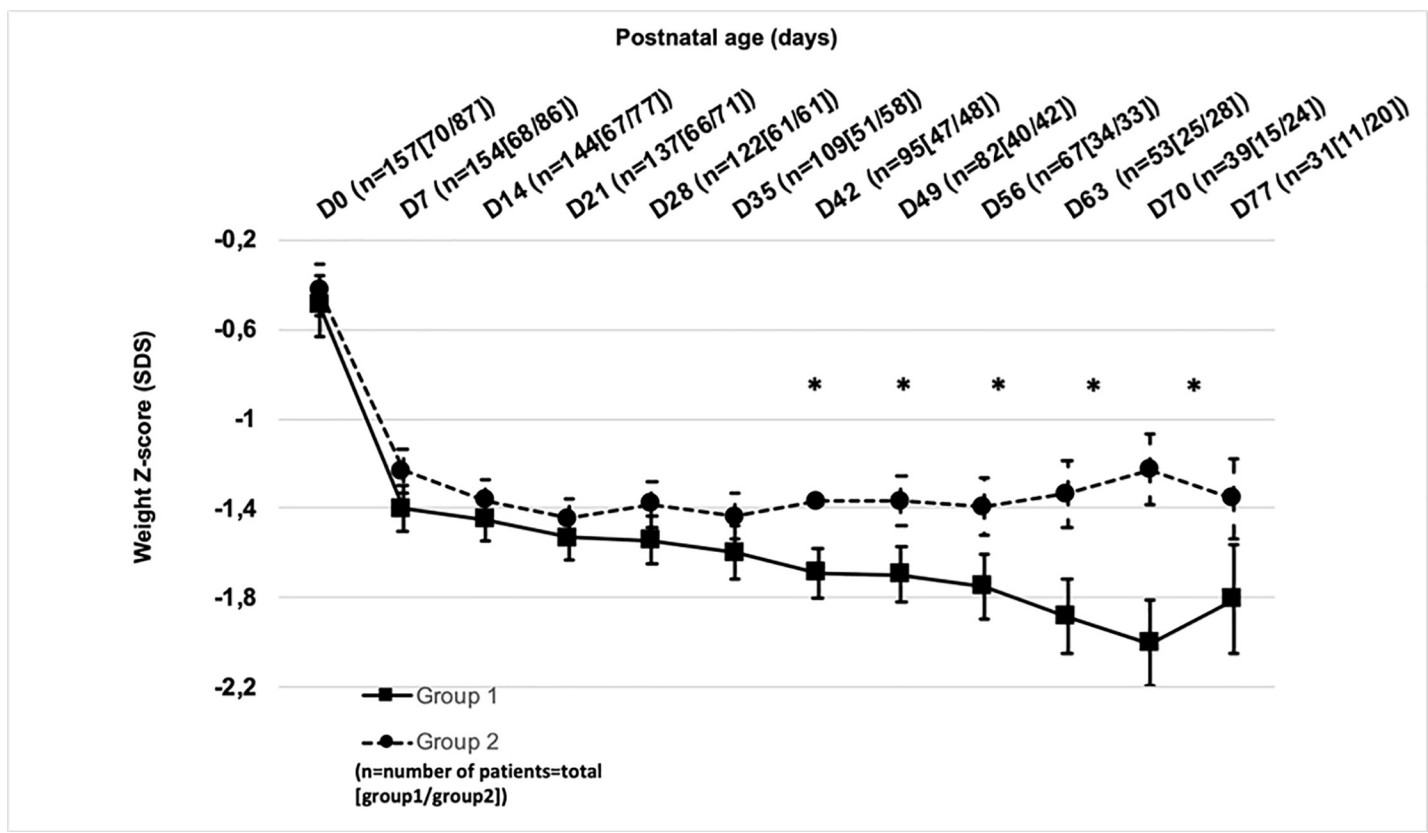

**Fig 1. Longitudinal weight z-score evolution from birth to day 77.** D = day, *p <0.05.

## Compliance to guidelines or standardized protocol

We observed low compliance to the protocol in both groups with regards to protein, lipid, and caloric intakes with a high variability. Fig 4 presents the nutrition data up to D42. There were no further differences between the groups from D42 onwards.

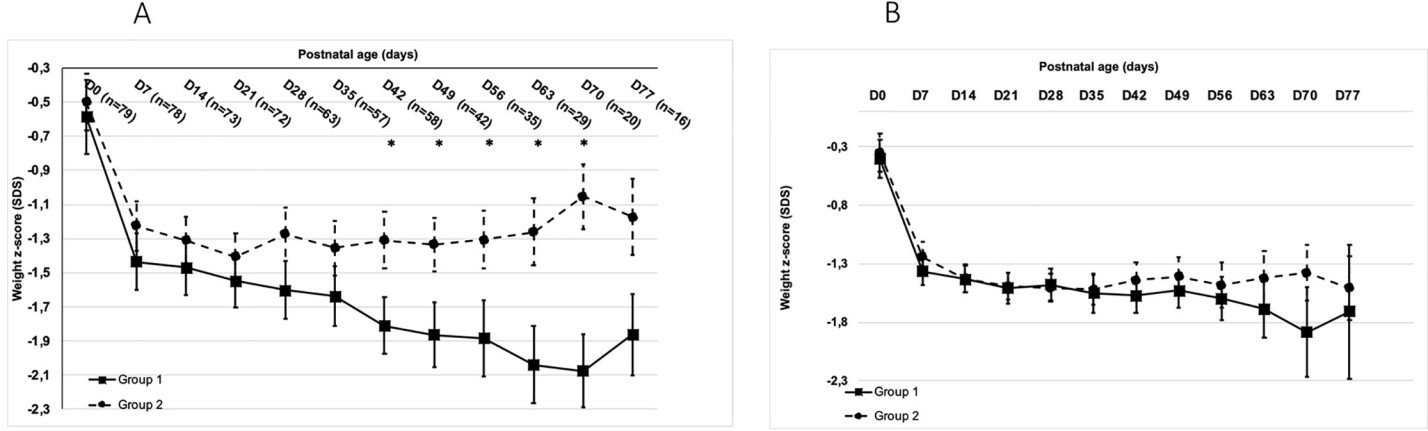

**Fig 2.** A: Longitudinal weight z-score evolution from birth to day 77 in males. D = day, *p <0.05. B: Longitudinal weight z-score evolution from birth to day 77 in females. D = day, p > 0.05 for all values.

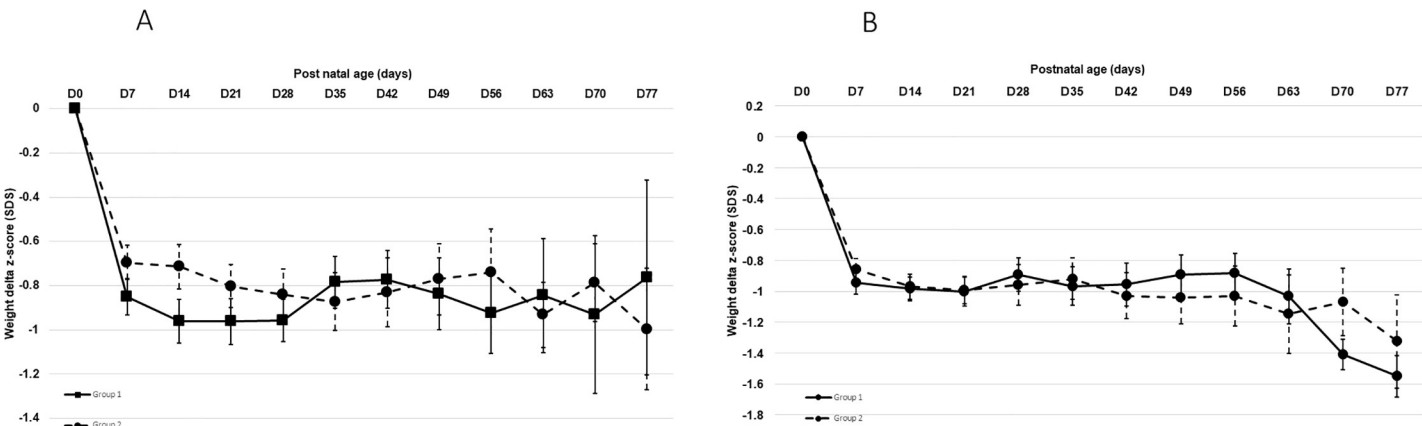

**Fig 3.** A: Longitudinal weight delta z-score evolution from birth to day 77 in males. D = day, p > 0.05 for all values. B: Longitudinal weight delta z-score evolution from birth to day 77 in males. D = day, p > 0.05 for all values.

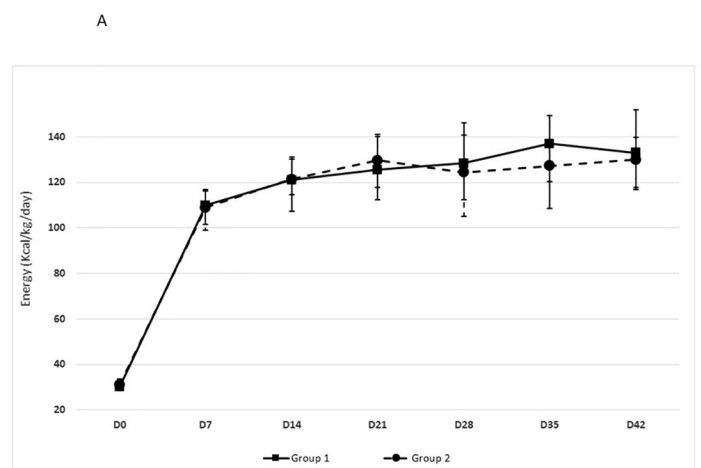

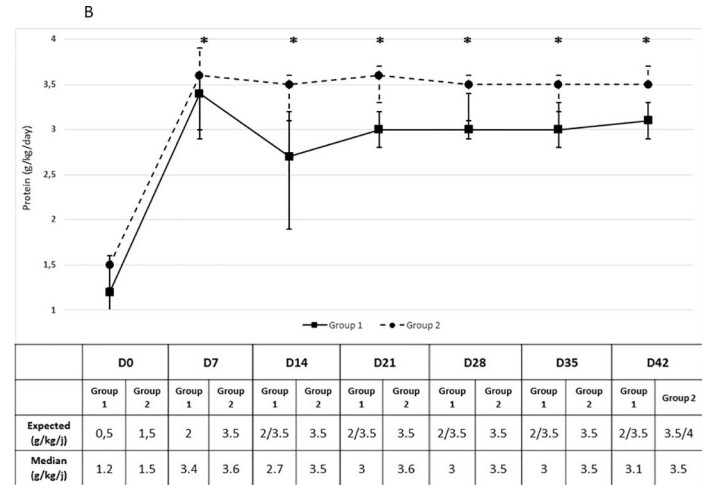

| | D0 | | D7 | | D14 | | D21 | | D28 | | D35 | | D42 | |
|---|---|---|---|---|---|---|---|---|---|---|---|---|---|---|
| | Group 1 | Group 2 | Group 1 | Group 2 | Group 1 | Group 2 | Group 1 | Group 2 | Group 1 | Group 2 | Group 1 | Group 2 | Group 1 | Group 2 |
| Expected (g/kg/j) | 0,5 | 1,5 | 2 | 3.5 | 2/3.5 | 3.5 | 2/3.5 | 3.5 | 2/3.5 | 3.5 | 2/3.5 | 3.5 | 2/3.5 | 3.5/4 |
| Median (g/kg/j) | 1.2 | 1.5 | 3.4 | 3.6 | 2.7 | 3.5 | 3 | 3.6 | 3 | 3.5 | 3 | 3.5 | 3.1 | 3.5 |

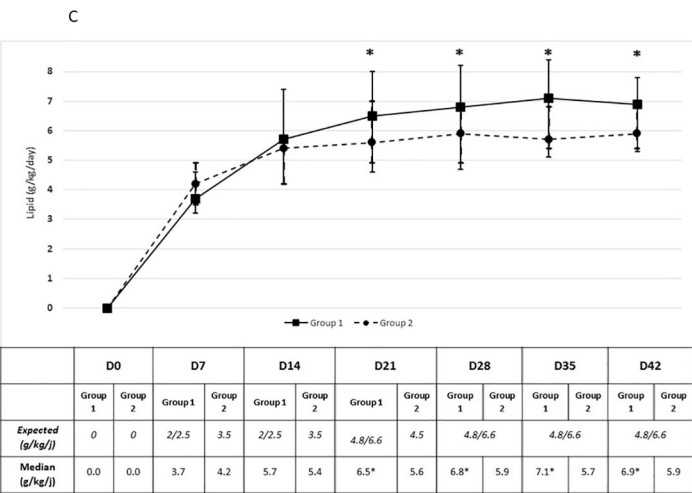

| | D0 | | D7 | | D14 | | D21 | | D28 | | D35 | | D42 | |
|---|---|---|---|---|---|---|---|---|---|---|---|---|---|---|
| | Group 1 | Group 2 | Group 1 | Group 2 | Group 1 | Group 2 | Group 1 | Group 2 | Group 1 | Group 2 | Group 1 | Group 2 | Group 1 | Group 2 |
| Expected (g/kg/j) | 0 | 0 | 2/2.5 | 3.5 | 2/2.5 | 3.5 | 4.8/6.6 | 4.5 | 4.8/6.6 | | 4.8/6.6 | | 4.8/6.6 | |
| Median (g/kg/j) | 0.0 | 0.0 | 3.7 | 4.2 | 5.7 | 5.4 | 6.5* | 5.6 | 6.8* | 5.9 | 7.1* | 5.7 | 6.9* | 5.9 |

**Fig 4. Median and variability (IQR) for (A) energy, (B) protein, and (C) lipid intakes from birth to day 42.** D = day, *p < 0.05.

Concerning caloric intakes, there was low compliance in both group throughout the observation period, but no significant differences in the median caloric intake between the 2 groups (Fig 4A). Compliance for caloric intake was 27.9%+/-19.8 in group 1 and 23%+/- 26.1% in group 2. Of note, Group 2 received a significantly higher protein intake and Group 1 received a significantly higher lipid intake from day 0 to day 42. The median protein intake varied from 1.2 to 3.1 g/kg/d at day 42 in Group 1 and from 1.5 to 3.5 g/kg/d at day 42 in Group 2, with significant differences between the two groups from day 7 to day 42 (Fig 4B and 4C). Compliance to guidelines was also low for protein intakes (53% +/-35.2 in group 1 and 38.6% +/- 26.4 in group 2), and for lipid intakes (26.9% +/- 25.6 in group 1 and 36.6% +/- 31 in group 2).

## Discussion

In this study, we showed that the introduction of a strict nutritional protocol, designed according to the most recent recommendations, led to a higher weight gain velocity from the sixth to the eleventh week of life with a limitation of PGR in very preterm infants. We observed a significantly attenuated drop in weight z-score compared with infants receiving nutrition based on the older recommendations. The difference was not statistically significant after 40 weeks PCA because the number of infants discharged from the hospital increased, thus the analysis lacked power. Our results also showed that the minimal z-score was higher and occurred earlier in Group 2 than in Group 1, with a catch-up effect after day 35 not observed in Group 1. These results suggest that a strictly implemented nutritional protocol may indeed improve growth in very preterm infants, limiting PGR severity before discharge Roggero et al.[12] studied 102 very low birth weight infants prospectively after the implementation of a new nutritional practice, compared to 69 infants from an historical cohort. They showed that discharged weight z-score was significantly higher in the intervention group (-1.7 vs 2.2 p = 0.001) which is consistent with our study. Likewise, Rochow et al. [13] introduced a set of evidence-based strategies in 123 premature infants, compared to 115 controls in a pre/post study design. They showed that optimization of early nutrition was associated to birth weight return 3 days earlier than the control group and an increased weight at 36 weeks PCA (delta = 260g, p<0.05) [13].

Of note, there was no difference in raw weight between the two groups, confirming that raw weight is not a good indicator for following infants' growth in daily practice. Indeed, weight z-score is a better predictor to diagnose growth restriction and adjust nutrition [24]. Using raw weight trajectory in 396 patients, after introduction of improved nutritional guidelines, Andrews et al. [11] found that the median change between birth and 36 weeks PCA was -0.27 SD score which is consistent with our findings.

Early nutritional deficits were not regained before hospital discharge in our study. This could be explained by poor compliance to the protocol. A significant part of PGR in preterm infants has been linked to nutritional deficit, which is mainly caused by fear of metabolic intolerance or NEC [2,3]. Looking at individual infant's files to verify their actual nutrition intake with regards to guidelines (Group 1) or the strict protocol (Group 2), we observed low compliance in both groups for the first month of life and throughout the observation period. This is consistent with Lapillonne et al [25]. Of note, we observed a significantly higher protein intake in Group 2, as expected when using the new protocol, but a significantly higher lipid intake in Group 1, with no differences in caloric intake between both groups. Maybe this low compliance was due to higher expectations, so despite higher intakes, compliance was not reached.

Thus, our two groups were demographically and clinically comparable, but their nutritional intake was qualitatively different. This point raises the question about the quality of growth in formerly premature infants [26]. We can speculate that more protein intake is better because growth velocity is increased in our study. Previously published studies confirm that earlier and

higher protein intake by preterm infants may increase weight gain velocity. Also they may achieve a leaner mass closer to that of full term infants [24]. We need to improve our practice and strictly follow the nutritional protocol. Early growth improvement is very important because there seems to be a window, in the first weeks of life, when interventions may have an impact [27,28]. We showed that despite appropriate energy intake and more protein intake, there is a catch-up growth but delayed by about one month. We speculate that lower protein intake may have a delayed maturation effect due to cumulative deficit during the first month [2].

Evaluating weight z-score, there was a significant growth improvement between the two periods for males only. This difference observed in growth trajectory is consistent with Christmann and Hack long term observations [29, 30]. However, this difference may also be due in part to the fact that males had lower weight z-scores at birth than females in our population. Therefore, we further analyzed the data accounting for the weight of the infants at birth using delta z-score from birth. We did not find significant differences anymore but delta z-score were stable with time and always above -1, without significant difference between sex but with a trend of a worse evolution in female from Day 56 (Fig 3B).

This suggests that the introduction of a nutrition protocol updated on the last guidelines had an impact on nutrient delivery, but the impact on growth was not what we expected because of low compliance to the protocol. In addition, as we worked on a cohort we might have had a lack of power.

We evaluated the effects of our new protocol on clinical outcomes other than growth and did not find any difference except a higher rate of surgery for PDA in Group 2, and longer doxapram treatment in Group 2. These differences may be explained by a better availability of pediatrics surgeons from 2014, and by a modification in our protocol for doxapram treatment between the two periods. Unfortunately, due to the retrospective design of our study, no data were available on respiratory support (mechanical ventilation and/or continuous positive airway pressure) to confirm this speculation.

The rate of NEC was lower in group 2 but not statistically different despite that enteral nutrition was increased faster using the new protocol. This is consistent with good tolerance of early "aggressive" nutrition for preterm infants [2].

The two groups were similar for the duration of central venous line and parenteral nutrition. We expected a shorter duration with the new protocol because enteral feeding was increased faster and the central venous line should be removed earlier than before (140 ml/kg/day of enteral feeding after *versus* 160 ml/kg/day before). This part of the protocol was obviously not fulfilled. These results demonstrate that when a new protocol is introduced in a unit, medical caregivers must be trained and that a time for adaptation is needed, as a recent study suggested it [31].

The strengths of the study include the extensive retrospective standardized collection of data as well as the routinely measured weight values throughout the study period.

Conversely, our study has limitations. It was not a randomized trial; however, it would not be ethical to voluntarily limit nutritional support in some infants. Confounding factors, such as medical treatment and mechanical ventilation could have affected the results, but were not investigated in this cohort study. However, we may speculate that a 6-month washout period was short enough to prevent significant modifications in our practice and allow for a comparison of both cohorts. Other limitations include the retrospective collection of nutritional data and a high dropout rate due to hospital discharge. Thus, we cannot exclude that the lack of difference observed from D70 onwards may be related to a lack of power. There is also variability in the protein and the fat composition of human milk that we were not able to determine at the time of the study [32].

## Conclusion

Feeding preterm neonates is extremely challenging. Optimal nutrition with a low incidence of complications requires a well-organized structure. Hospital staff must acknowledge that preterm birth is also a nutritional emergency which could result in serious short- and long-term detrimental effects. Our data suggest that the quality of nutritional care using a strictly-defined protocol may improve weight gain for very preterm infants. The results are promising since this policy was able to limit PGR at discharge.

This study was based on an internal audit and we noted that what we think we are doing is not always what we are really doing. That kind of evaluation may be recommended to all units to point out what improvements can be made for optimizing nutritional care for these vulnerable infants.

## Supporting information

**S1 Data.**
(XLS)

**S2 Data.**
(XLS)

**S3 Data.**
(XLSX)

**S4 Data.**
(XLSX)

**S5 Data.**
(XLS)

## Author Contributions

**Conceptualization:** Apolline Wittwer, Jean-Michel Hascoët.

**Data curation:** Apolline Wittwer.

**Formal analysis:** Apolline Wittwer, Jean-Michel Hascoët.

**Methodology:** Apolline Wittwer, Jean-Michel Hascoët.

**Project administration:** Apolline Wittwer.

**Resources:** Apolline Wittwer.

**Supervision:** Jean-Michel Hascoët.

**Validation:** Jean-Michel Hascoët.

**Visualization:** Apolline Wittwer, Jean-Michel Hascoët.

**Writing – original draft:** Apolline Wittwer.

**Writing – review & editing:** Apolline Wittwer, Jean-Michel Hascoët.

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
