## [Decision Letter · Decision Letter 0]

7 Feb 2020

PONE-D-19-35465

Impact of introducing a standardized nutrition protocol on very premature infants’ growth and morbidity

PLOS ONE

Dear Dr. WITTWER,

Thank you for submitting your manuscript to PLOS ONE. After careful consideration, we feel that it has merit but does not fully meet PLOS ONE’s publication criteria as it currently stands. Therefore, we invite you to submit a revised version of the manuscript that addresses the points raised during the review process.

I kindly ask you to address all points raised by both reviewers in a revised version of the ms including a detailled point-by-point response to each comment addressing the changes in the revised version. Please pay special attention to the SDS and z-score discrepancies and the proper and detailled inclusion of congruencies and disparities of the presented results in comparison to the published literature. 

We would appreciate receiving your revised manuscript by Mar 23 2020 11:59PM. To enhance the reproducibility of your results, we recommend that if applicable you deposit your laboratory protocols in protocols.io, where a protocol can be assigned its own identifier (DOI) such that it can be cited independently in the future. For instructions see: http://journals.plos.org/plosone/s/submission-guidelines#loc-laboratory-protocols

We look forward to receiving your revised manuscript.

Kind regards,

Harald Ehrhardt

Academic Editor

PLOS ONE

Journal Requirements:

2. Please include your tables as part of your main manuscript and remove the individual files. Please note that supplementary tables (should remain/ be uploaded) as separate "supporting information" files

Reviewers' comments:

Reviewer's Responses to Questions

**Comments to the Author**

1. Is the manuscript technically sound, and do the data support the conclusions?

Reviewer #1: Partly

Reviewer #2: Yes

2. Has the statistical analysis been performed appropriately and rigorously? 

Reviewer #1: No

Reviewer #2: Yes

3. Have the authors made all data underlying the findings in their manuscript fully available?

Reviewer #1: Yes

Reviewer #2: Yes

4. Is the manuscript presented in an intelligible fashion and written in standard English?

Reviewer #1: Yes

Reviewer #2: Yes

5. Review Comments to the Author

Reviewer #1: This paper has been improved by the responses the authors have made to the reviewer comments, and I am appreciative of their efforts. However, there are a few more issues which would improve the paper further if they were to be addressed.

The main issue is that, whilst they have now added in something about the changes in SDS from birth in their groups (line 139-140), they have not shown this data properly, nor integrated well into the paper. It seems there were not significant differences between the groups with regard to change in SDS from birth, and this is an important finding. As change in SDS from birth takes into account the differences in birth weight in each group, the fact that the authors could not demonstrate a significant improvement in the fall in SDS from birth to each time point, means that their intervention perhaps was not as effective at improving growth as they suggest it is. The change in SSD is more meaningful that raw SDS, so for me this non-significant difference supersedes the significant findings between the two groups in terms of raw z score. The authors need to better integrate and acknowledge the new results in to the paper (as a table or figure) and need to comment more on this non-significant finding in the results. This is not to say that the work here is not important- they demonstrated that their intervention had a significant impact on nutrient delivery, and this is important. The fact it doesn’t to appear to have made an impact on growth may fit with their finding that compliance with the new practices were poor, and is again important and needs further thought and comment. In relation to this, the authors compared group using t-tests or mann-whitney tests. If the growth and nutrition data is normally distributed, then they should use linear regression where possible rather than t tests, as this would enable them to adjust for baseline factors between groups such as sex, gestational age and weight at birth. This may also impact on their results

My other main comment, is that whilst the authors have now included some pertinent references of similar studies (Andrews et al. Rogerro et al and Rochow et al), they have only included these in the introduction and not the discussion, which needs to be addressed. In the discussion, the authors need to comment on how their findings are similar to or differ from these earlier studies, and where there are differences, what are they and why do the authors think this occurred?

There other some other, more specific, minor issues which need to be addressed:

Abstract- this is fine though the results section is packed with dense figures and is a little hard to read, The authors need to make their headlines clear here

Line 73- this should read SDS and not DS. Also, WOG (week of gestation) would be better replaced with the term Post-conceptual Age (PCA). Figures 1 and 2 should also use SDS and not DS in the y axis labels

Line 136- the authors still use the term ‘significantly more important’ and I’m not sure what they mean by this. Looking at the results I think they mean ‘significantly smaller’, but this needs to be amended

Reviewer #2: The authors have addressed most of my comments as far as available.

As with many retrospective studies some relevant data are missing which is addressed in the discussion.

I have a few minor comments to be addressed.

Page 11 line 209 I think it should read We speculate that lower protein intake may have a delayed maturation effect......

Page 11 line 218. I would add a sentence here that unfortunately there were no data available on respiratory support (echanical ventilation and/or CPAP). I expect that that would be the most important factor why longer doxapram was given?

The references are still inconsistent including some french words and eg ref 33 is incomplete.

6. PLOS authors have the option to publish the peer review history of their article (what does this mean?). If published, this will include your full peer review and any attached files.

Reviewer #1: No

Reviewer #2: No

---

## [Author Response · Author response to Decision Letter 0]

26 Mar 2020

Please include your tables as part of your main manuscript and remove the individual files. Please note that supplementary tables (should remain/ be uploaded) as separate "supporting information" files

As requested, we included the tables as part of the main manuscript

Reviewers' comments:

Reviewer's Responses to Questions

Comments to the Author

1. Is the manuscript technically sound, and do the data support the conclusions?

Reviewer #1: Partly

Reviewer #2: Yes

2. Has the statistical analysis been performed appropriately and rigorously?

Reviewer #1: No

Reviewer #2: Yes

3. Have the authors made all data underlying the findings in their manuscript fully available?

The PLOS Data policyhttp://www.plosone.org/static/policies.action#sharing requires authors to make all data underlying the findings described in their manuscript fully available without restriction, with rare exception (please refer to the Data Availability Statement in the manuscript PDF file). The data should be provided as part of the manuscript or its supporting information, or deposited to a public repository. For example, in addition to summary statistics, the data points behind means, medians and variance measures should be available. If there are restrictions on publicly sharing data—e.g. participant privacy or use of data from a third party—those must be specified.

Reviewer #1: Yes

Reviewer #2: Yes

4. Is the manuscript presented in an intelligible fashion and written in standard English?

Reviewer #1: Yes

Reviewer #2: Yes

5. Review Comments to the Author

Reviewer #1: This paper has been improved by the responses the authors have made to the reviewer comments, and I am appreciative of their efforts. However, there are a few more issues which would improve the paper further if they were to be addressed.

The main issue is that, whilst they have now added in something about the changes in SDS from birth in their groups (line 139-140), they have not shown this data properly, nor integrated well into the paper. It seems there were not significant differences between the groups with regard to change in SDS from birth, and this is an important finding. As change in SDS from birth takes into account the differences in birth weight in each group, the fact that the authors could not demonstrate a significant improvement in the fall in SDS from birth to each time point, means that their intervention perhaps was not as effective at improving growth as they suggest it is. The change in SSD is more meaningful that raw SDS, so for me this non-significant difference supersedes the significant findings between the two groups in terms of raw z score. The authors need to better integrate and acknowledge the new results in to the paper (as a table or figure) and need to comment more on this non-significant finding in the results. This is not to say that the work here is not important- they demonstrated that their intervention had a significant impact on nutrient delivery, and this is important. The fact it doesn’t to appear to have made an impact on growth may fit with their finding that compliance with the new practices were poor, and is again important and needs further thought and comment. 

As requested, we added in the Results section additional Figures (Fig 3A for boys and Fig.3B for girls) to illustrate the delta z-score analysis; 

In addition, we developed the comments on this non-significant finding in the discussion.

In relation to this, the authors compared group using t-tests or Mann-Whitney tests. If the growth and nutrition data is normally distributed, then they should use linear regression where possible rather than t tests, as this would enable them to adjust for baseline factors between groups such as sex, gestational age and weight at birth. This may also impact on their results

As stated in the Method section, we tested the normality of the data with a Shapiro-Wilk test of normality. Unfortunately, the fit is poor for several variables and significant for one of them indicated a lack of normal distribution. Therefore it is not appropriate to perform a linear regression for these data. In addition, the difference observed between each time point is rather low which allow us to speculate that a linear regression would not show any significant correlation. 

Using weight z-score adjusted for sex may be considered as a substitute for the impact of sex, gestational age and post-natal age. The analysis of delta weight z-score from birth should cope for weight at birth.

My other main comment, is that whilst the authors have now included some pertinent references of similar studies (Andrews et al. Rogerro et al and Rochow et al), they have only included these in the introduction and not the discussion, which needs to be addressed. In the discussion, the authors need to comment on how their findings are similar to or differ from these earlier studies, and where there are differences, what are they and why do the authors think this occurred?

As requested, we developed the 3 studies whose findings are consistent with our own data in the discussion section of the manuscript.

There other some other, more specific, minor issues which need to be addressed:

Abstract- this is fine though the results section is packed with dense figures and is a little hard to read, The authors need to make their headlines clear here

Line 73- this should read SDS and not DS. Also, WOG (week of gestation) would be better replaced with the term Post-conceptual Age (PCA). Figures 1 and 2 should also use SDS and not DS in the y axis labels

Line 136- the authors still use the term ‘significantly more important’ and I’m not sure what they mean by this. Looking at the results I think they mean ‘significantly smaller’, but this needs to be amended

All these points have been corrected.

Reviewer #2: The authors have addressed most of my comments as far as available.

As with many retrospective studies some relevant data are missing which is addressed in the discussion.

I have a few minor comments to be addressed.

Page 11 line 209 I think it should read We speculate that lower protein intake may have a delayed maturation effect......

Page 11 line 218. I would add a sentence here that unfortunately there were no data available on respiratory support (echanical ventilation and/or CPAP). I expect that that would be the most important factor why longer doxapram was given?

All these points have been corrected and a sentence added in the discussion as requested

The references are still inconsistent including some french words and eg ref 33 is incomplete.

We thoroughly revised the referent format to make it consistent throughout the manuscript. We corrected and completed the incomplete reference. We apologize for that problem.

6. PLOS authors have the option to publish the peer review history of their article (what does this mean?https://journals.plos.org/plosone/s/editorial-and-peer-review-process#loc-peer-review-history). If published, this will include your full peer review and any attached files.

Do you want your identity to be public for this peer review? For information about this choice, including consent withdrawal, please see our Privacy Policyhttps://www.plos.org/privacy-policy.

Reviewer #1: No

Reviewer #2: No

---

## [Decision Letter · Decision Letter 1]

21 Apr 2020

Impact of introducing a standardized nutrition protocol on very premature infants’ growth and morbidity

PONE-D-19-35465R1

Dear Dr. WITTWER,

We are pleased to inform you that you have carefully addressed all items raised and that your manuscript has been judged scientifically suitable for publication and will be formally accepted for publication once it complies with all outstanding technical requirements.

With kind regards,

Harald Ehrhardt

Academic Editor

PLOS ONE

Additional Editor Comments (optional):

Reviewers' comments:

Reviewer's Responses to Questions

**Comments to the Author**

1. If the authors have adequately addressed your comments raised in a previous round of review and you feel that this manuscript is now acceptable for publication, you may indicate that here to bypass the “Comments to the Author” section, enter your conflict of interest statement in the “Confidential to Editor” section, and submit your "Accept" recommendation.

Reviewer #1: All comments have been addressed

Reviewer #2: All comments have been addressed

2. Is the manuscript technically sound, and do the data support the conclusions?

Reviewer #1: Yes

Reviewer #2: Yes

3. Has the statistical analysis been performed appropriately and rigorously? 

Reviewer #1: Yes

Reviewer #2: Yes

4. Have the authors made all data underlying the findings in their manuscript fully available?

Reviewer #1: Yes

Reviewer #2: Yes

5. Is the manuscript presented in an intelligible fashion and written in standard English?

Reviewer #1: Yes

Reviewer #2: Yes

6. Review Comments to the Author

Reviewer #1: (No Response)

Reviewer #2: The reviewers have addressed my remaining comments.

I have no remaining comments except for the legend for figure 3A and B is now missing

7. PLOS authors have the option to publish the peer review history of their article (what does this mean?). If published, this will include your full peer review and any attached files.

Reviewer #1: No

Reviewer #2: No

---

## [Editor Report · Acceptance letter]

5 May 2020

PONE-D-19-35465R1 

Impact of introducing a standardized nutrition protocol on very premature infants’ growth and morbidity 

Dear Dr. Wittwer:

I am pleased to inform you that your manuscript has been deemed suitable for publication in PLOS ONE. Congratulations! Your manuscript is now with our production department. 

With kind regards,

on behalf of

Dr. Harald Ehrhardt 

Academic Editor

PLOS ONE